# Radiosurgery for Hypothalamic Gliomas: A Case Report and Clinical Guidelines Form a Neurosurgical Center of Excellence

**DOI:** 10.3390/jpm14111108

**Published:** 2024-11-15

**Authors:** Jose Valerio, Matteo Borro, Noe Santiago Rea, Laurel Valente, Maria P. Fernandez-Gomez, Jorge Zumaeta, Penelope Mantilla, Andres M. Alvarez-Pinzon

**Affiliations:** 1Neurosurgery Department, Palmetto General Hospital, Hialeah, FL 33313, USA; jevalerio@jvaleriomd.com; 2Neurosurgery Oncology Center of Excellence, Neurosurgery Department, Miami Neuroscience Center at Larkin, South Miami, FL 33143, USA; 3Neurosurgery Department, Latino America Valerio Foundation, Weston, FL 33331, USA; matteo.borro@hsanmartino.it (M.B.); nsantiago@jvaleriomd.com (N.S.R.); valente.l@northeastern.edu (L.V.); research@latinoamericavaleriofoundation.org (M.P.F.-G.); jzumaeta@jvaleriomed.com (J.Z.); pmantilla@jevaleriomd.com (P.M.); 4Internal Medicine Unit, Department of Internal Medicine, IRCCS Ospedale Policlinico San Martino, Largo R. Benzi 10, 16132 Genova, Italy; 5College of Science, Northeastern University, Boston, MA 02115, USA; 6Institute for Human Health and Disease Intervention, Division of Research, FAU Charles E. Schmidt College of Medicine, Florida Atlantic University, Boca Raton, FL 33431, USA; 7Cancer Neuroscience Program, The Institute of Neuroscience of Castilla y León (INCYL), Universidad de Salamanca, 37007 Salamanca, Spain

**Keywords:** glioma, astrocytoma, hypothalamus, radiosurgery, brain tumors, hypothalamic gliomas, cancer neuroscience

## Abstract

**Background:** Hypothalamic gliomas, though rare, present significant challenges in neuro-oncology due to their critical location and high morbidity risk. Pilocytic astrocytoma is the most common subtype, requiring a delicate balance between tumor control and preservation of neurological function. This study explores radiosurgery as a viable treatment option for hypothalamic gliomas, with a focus on low-grade lesions. **Methods:** A comprehensive literature review was conducted using the PubMed database to compare the outcomes of surgical and non-surgical interventions for hypothalamic gliomas. The purpose of case report and clinical guidelines is to present a case report and critically compare its outcomes with the data identified in the literature. We report the case of a 25-year-old male presenting with hypernatremia, diabetes insipidus, and altered mental status. Imaging revealed a hypothalamic lesion consistent clinically with a low-grade astrocytoma. Given the tumor’s proximity to critical neurovascular structures, gamma knife radiosurgery was chosen as the intervention. Post treatment, tumor reduction and neurological improvement were observed and documented. **Results:** The case demonstrated a significant decrease in tumor size and resolution of neurological symptoms post radiosurgery. The management of hypothalamic gliomas remains contentious due to the complex anatomy of the hypothalamus. While surgical resection remains common, it carries a high risk of morbidity. Radiosurgery offers a minimally invasive alternative, effectively controlling tumor growth while reducing neurological risk. The favorable outcomes in our case, supported by the literature, highlight the efficacy of radiosurgery, particularly for low-grade astrocytomas. **Conclusions:** Gamma knife radiosurgery presents a promising alternative to conventional surgery for hypothalamic gliomas, particularly low-grade lesions such as pilocytic astrocytomas. The reduction in surgical risk and positive clinical outcomes make radiosurgery a valuable tool in the management of these challenging tumors.

## 1. Introduction

Hypothalamic gliomas have been reported with optic pathway gliomas and are infrequently reported as a different entity [1]. Because it is difficult to differentiate between optic chiasm gliomas and hypothalamic gliomas, both radiologically and clinically, these two entities tend to be relegated under the heading of optic pathway gliomas. Hypothalamic gliomas constitute one-third of the optic pathway tumors [2].

Optic pathway gliomas make up 2–5% of brain tumors in children [3,4]. These tumors usually affect the visual pathway before reaching the cortex and can spread from the optic nerves to the occipital lobe cortex, also involving the area around the hypothalamus and optic chiasm [3]. Hypothalamic gliomas (HGs) have been described as rare pilocytic low-grade astrocytomas, tend to be benign tumors with loose stroma and multicystic degeneration, are usually encompassed among parasellar lesions, and are more common among children than adults [1,5,6,7,8]. In adults, HG accounts for 9–30% of cases, especially in patients who have neurofibromatosis type 1 [9]. In the pediatric population, HG accounts for 10–15% of supratentorial tumors [7].

Despite the era of microsurgery, and significant debulking results, the hypothalamus, and vital anatomical structures, such as the dopaminergic pathway and the hypothalamic-hypophyseal axis, are at risk of injury [8].

The purpose of this manuscript is to describe a case of a hypothalamic lesion consistent with pilocytic astrocytoma. Instead of using conventional microsurgery, the lesion was treated with radiosurgery, which resulted in positive outcomes without increasing the patient’s risk of complications or mortality, as compared to biopsy or partial tumor removal.

## 2. Materials and Methods

The study protocol (LCH-8-062021) received approval from the Institutional Review Board under the “CNS, Neurosurgery Oncology and Vascular Neurology Outcomes Research” protocol (Last approval: 1 November 2023). Data collection and analysis adhered to institutional and ethical standards. This case report and clinical guideline aim to provide a detailed presentation of a unique hypothalamic glioma case and critically evaluate its outcomes in comparison with existing literature. Comprehensive clinical data, including patient presentation, imaging, treatment approach, and outcomes, were systematically collected and meticulously recorded to ensure accuracy. This analysis informed the development of detailed management guidelines for hypothalamic gliomas, emphasizing precision in reporting and clinical relevance.

The PubMed database was employed to conduct a systematic literature review that focused on publications published between 1994 and 2024. A combination of relevant keywords, such as “Hypothalamic Gliomas”, “Radiosurgery”, “Astrocytoma”, and “Hypothalamus”, was implemented in the search strategy. A total of 30 pertinent articles were identified and evaluated. The review primarily concentrated on publications that addressed hypothalamic gliomas. However, studies that discussed both hypothalamic and optic pathway gliomas were included when they were appropriate, as the two entities share anatomical and pathological similarities.

Data extraction was followed by a critical analysis and synthesis of the findings, with a particular focus on the clinical presentation, imaging features, and treatment approaches for hypothalamic gliomas. The results were subsequently positioned within the broader context of current clinical practice and the role of radiosurgery in the treatment of hypothalamic gliomas through a comparative analysis of the case report and the literature. The objective was to assess the efficacy of radiosurgery and its clinical outcomes in comparison to current treatment modalities.

## 3. Results

A 25-year-old male with no significant prior medical history presented to the emergency department with a cognitively altered state that had been progressively deteriorating over the course of several weeks. In the initial assessment, the patient exhibited signs of lethargy and confusion. In the suprasellar region, posterior to the optic chiasm, a well-defined, hyperdense mass was identified by a non-contrast computed tomography (CT) scan of the brain. No calcifications were observed in the lesion. After the CT findings, the patient developed symptoms that were indicative of central diabetes insipidus, including hypernatremia and polyuria (serum osmolality of 349 mOsm/kg, sodium level of 164 mmol/L, and creatinine level of 2.6 mg/dL).

The patient’s neurological examination was characterized by encephalopathy and lethargy. Neurological reflexes were bilaterally graded 2+ and symmetrical in both the upper and lower extremities. Withdrawal responses to painful stimuli were observed; however, no other focal deficits were revealed. Given the mass’s suprasellar location and its proximity to critical neurovascular structures, additional assessment was necessary.

The MRI was subsequently conducted using a navigation protocol, which revealed a well-circumscribed lesion in the hypothalamic region that was clinically and anatomically consistent with a low-grade astrocytoma. The optic chiasm, pituitary stalk, and adjacent neurovascular structures were all in close proximity to the tumor, which presented substantial risks if conventional surgical intervention were pursued.

Due to the clinical and radiological diagnosis of a pilocytic astrocytoma, classified as a low-grade astrocytoma (WHO Grade I), the primary treatment selected was gamma knife radiosurgery (GKRS). This decision followed extensive multidisciplinary discussions and thorough consultation with the patient’s surrogate (Figure 1). The high-risk anatomical location of the hypothalamic lesion, where traditional open surgical resection posed significant morbidity risks—such as damage to critical neurovascular structures—was a key factor. Given the lesion’s proximity to the optic pathways, pituitary gland, and other vital neuroendocrine structures, GKRS was deemed the most appropriate intervention due to its precision and minimally invasive approach, offering targeted therapy with a reduced risk of neurological deficits (Table 1).

Before proceeding with radiosurgery, the patient’s diabetes insipidus was medically managed to stabilize the electrolyte imbalance, which was essential to optimizing the patient’s condition for the procedure. Steroid therapy was considered but not required due to the absence of significant perilesional edema.

GKRS was performed using advanced stereotactic navigation, which enabled precise delineation of the tumor margins while minimizing radiation exposure to the surrounding healthy tissues (Table 2). The radiosurgical plan was carefully developed to provide optimal dose coverage to the lesion while adhering to dose constraints for critical structures, particularly the optic chiasm, brainstem, and hypothalamus. A marginal dose of 14 Gy was delivered to the 50% isodose line, which provided an effective tumoricidal dose with minimal risk of radiation-induced complications. The precision afforded by stereotactic targeting was integral to achieving the therapeutic objectives, given the high sensitivity of the surrounding anatomy to radiation.

The patient tolerated the procedure well, with no immediate post-procedural complications noted (Table 3). The absence of acute adverse effects, such as edema or neurological deterioration, was indicative of both the accuracy of the radiosurgical targeting and the careful pre-procedural planning. Follow-up imaging demonstrated partial tumor reduction, and early clinical assessments revealed an improvement in the patient’s neurological status, including stabilization of diabetes insipidus and resolution of hypernatremia.

At the three-month follow-up following GKRS, the patient exhibited substantial clinical improvement (Figure 2). He no longer demonstrated symptoms of encephalopathy or lethargy, and his mental status had returned to “normal.” Furthermore, the patient’s neurological examination was unremarkable, and repeat MRI imaging demonstrated a significant decrease in tumor volume. Desmopressin therapy maintained satisfactory control of diabetes insipidus, and no new neurological deficits were identified (Figure 3). The efficacy of radiosurgery as a minimally invasive treatment option for hypothalamic gliomas is emphasized by these results, particularly in situations where conventional surgery demonstrates significant risks.

Description: The table provides a summary of the data collected during a stereotactic radiosurgery session to treat a glioma. A maximum dose of 22.0 Gy was delivered to the tumor target, with a total of 9 radiation shots administered. The prescribed dose was 11.0 Gy, which was delivered to the 50% isodose line. To guarantee high accuracy in dose delivery, stereotactic coordinates (X = 96.5 mm, Y = 108.5 mm, and Z = 107.0 mm) were employed to achieve precise localization with a spatial resolution of 1.0 mm. The treatment was carefully planned to optimize tumor control while minimizing radiation exposure to adjacent healthy structures, as evidenced by the total beam-on time of 109.6 min.

Table 2: This table details “Run 1 (90 degrees)” in a stereotactic radiation session, with six shots targeting specific coordinates (X: 94.7–103.9 mm, Y: 101.2–114.6 mm, Z: 101.3–112.1 mm). Collimator settings mainly used position 8, with selective use of position B. Shot durations ranged from 8.89 to 14.12 min, reflecting precise, tailored dose delivery to the target area.

Table 3: This table provides a detailed description of “Run 2 (110 degrees)” in a stereotactic radiation session, which involves three shots aimed at precise coordinates in the X, Y, and Z planes. The collimator settings predominantly utilize position 8, with occasional utilization of position B. The shot durations vary between 12.25 and 14.85 min, demonstrating accurate and tailored administration of the dosage to the specific target areas.

## 4. Discussion

Pilocytic astrocytomas, classified as low-grade astrocytomas (WHO Grade I), are typically managed through surgical resection, particularly when the tumor is symptomatic or demonstrating progression. Complete resection, when feasible, remains the gold standard and is associated with favorable long-term outcomes. However, anatomical constraints, especially in eloquent regions such as the hypothalamus, frequently limit the ability to achieve gross total resection. In such cases, partial resection or biopsy is performed to establish a diagnosis and alleviate symptoms, with alternative management strategies including observation, radiotherapy, or chemotherapy tailored to the patient’s clinical and radiographic progression.

In pediatric populations, the use of radiotherapy is generally avoided due to its well-documented long-term adverse effects, including impaired cognitive development, neuroendocrine dysfunction, and an increased risk of radiation-induced secondary malignancies. The developing brain is particularly sensitive to radiation, and thus, radiotherapy is reserved for cases of recurrent or progressive disease where other treatment options are limited. In adult patients, however, radiotherapy is more readily considered as a viable therapeutic option, especially when surgical resection is contraindicated or incomplete.

Advanced radiation techniques, such as stereotactic radiosurgery (SRS) and stereotactic radiation therapy (SRT), provide highly targeted radiation delivery to the tumor while minimizing exposure to surrounding healthy tissues. These modalities reduce the potential for radiation-induced side effects and have demonstrated efficacy in controlling tumor progression. SRS and SRT are increasingly recognized as valuable tools in the management of hypothalamic pilocytic astrocytomas, offering a non-invasive approach to tumor control in both pediatric and adult populations where traditional surgery may pose significant risks.

In cases where hypothalamic involvement precludes safe surgical intervention, these advanced radiotherapeutic techniques serve as critical components of a multimodal treatment strategy. Their precision in targeting tumor tissue, while sparing adjacent neurovascular structures, underscores their utility in managing progression in anatomically sensitive regions like the hypothalamus [10].

### 4.1. Hypothalamic Gliomas—Radiologic Characteristics

Hypothalamic gliomas differentiate from optic gliomas such that they are seen as large masses in the suprasellar region infiltrating the brain parenchyma and third ventricle, while optic gliomas infiltrate along the optic pathway [11]. Hypothalamic gliomas typically present as rounded or lobulated solid masses located above the Sella turcica [12]. Hypothalamic gliomas appear hypointense on T1-weighted sequences and hyperintense on T2-weighted [12,13] and FLAIR sequences, with larger tumors being heterogeneous and containing both cystic and solid components. The solid component exhibits significant contrast enhancement [13]. The enhancement pattern can change over time without any clinical significance. If the tumor extends upwards and compresses the third ventricle and the foramen of Monro, hydrocephalus might result. Infiltration of the subarachnoid spaces might occur, showing brightness on T2-weighted images and a ring of enhancement [12]. The presumptive diagnosis of our case was pilocytic astrocytoma due to the presence of a hypothalamic multilobulated lesion that enhanced with contrast with an associated cystic lesion. Hypothalamic hamartoma was ruled out since contrast enhancement was evident, likewise, the clinical manifestations of our case were different from gelastic seizures [14,15]. Craniopharyngioma diagnosis was not considered due to the absence of calcifications associated with the tumor and the cyst component was not hyperintense in T1 [16].

One characteristic of hypothalamic gliomas is that they do not destroy the hypothalamus; instead, this tumoral mass only displaces and compresses it. However, due to the slow growth of hypothalamic gliomas, clinical symptoms and signs occur minimally. Hypothalamic gliomas are known as biologically indolent [1], and their common manifestation is diencephalic syndrome when presented during infancy [1,2]; likewise, it affects vision and can cause growth hormone dysfunction [7]. Less frequent symptoms such as precocious puberty, hypersomnia, and appetite disturbances such as anorexia or obesity can be manifested [2]. During acute presentation, hemorrhage might be possible with a subsequent obstructive hydrocephalus [6]. Considering the above, our case is interesting since our patient’s clinical symptoms differ from what is classically described in the literature. Our patient presented with altered mental status which progressed, requiring hospitalization and imaging studies, evidencing a hypothalamic tumoral mass.

### 4.2. Prognostic Factors

Prognostic factors in low-grade hypothalamic gliomas differ significantly from those in low-grade supratentorial gliomas due to the unique challenges posed by their location and the involvement of critical neurovascular structures. In hypothalamic gliomas, the primary factors influencing prognosis include tumor size, the extent of resection, patient age at diagnosis, and the presence of neuroendocrine dysfunction, such as diabetes insipidus or hypothalamic–pituitary axis abnormalities. Complete resection is often limited due to the high-risk anatomical location, and this contributes to lower progression-free survival (PFS) and overall survival (OS) rates compared to low-grade gliomas in more surgically accessible regions [3,4,5,6,7].

Age has been consistently identified as an important prognostic factor, with pediatric patients generally showing better outcomes than adults, likely due to the more indolent nature of pilocytic astrocytomas in younger populations. Additionally, the presence of neuroendocrine symptoms prior to diagnosis has been associated with poorer outcomes, as these symptoms often reflect more extensive tumor involvement in the hypothalamus and surrounding structures [8,9,10,11].

In contrast, prognostic factors in low-grade supratentorial gliomas include the tumor’s molecular profile, such as the presence of IDH mutations and 1p/19q codeletions, which are strongly associated with better outcomes and longer survival. These molecular markers are less commonly identified in hypothalamic gliomas, suggesting a distinct biological behavior. In supratentorial gliomas, complete surgical resection remains a primary predictor of improved survival, as gross total resection is more frequently achievable [12,13,14,15,16,17,18,19].

Radiotherapy and chemotherapy are considered as adjunct therapies in both hypothalamic and supratentorial gliomas, particularly in cases where complete resection is not feasible [20,21,22,23,24,25]. However, the role of these therapies in hypothalamic gliomas, especially in pediatric populations, is more limited due to the potential for long-term cognitive and neuroendocrine side effects [25,26,27,28,29,30,31,32].

In a recent study by Deacu et al. [32], prognostic factors for low-grade gliomas were evaluated, showing that while factors such as age, molecular profile, and extent of resection play a significant role in supratentorial gliomas, the unique challenges of hypothalamic gliomas require a more conservative approach focused on preserving neurological function. The comparison highlights the necessity of individualized treatment strategies, particularly in hypothalamic gliomas where surgical limitations and neuroendocrine complications are paramount in determining outcomes [32,33,34,35].

### 4.3. GKRS in Hypothalamic Gliomas

Some glioma cells are hypoxic, leading them to be resistant to damage by radiation. When these cells are irradiated, and sublethal insults accumulate at low doses, this imparts a cumulative effect on their life cycle. Several theories have been postulated to explain this, but apparently, DNA is the main target for cellular damage from ionizing radiation, leading to double-stranded breaks that trigger cell cycle arrest and cell death. A single dose of radiation therapy can reduce cell viability by as much as 2.5 to 3 times. Oxygenated cells are much more sensitive to radiation compared to hypoxic cells and primarily determine the overall response to radiation. GKRS capitalizes on the natural differences in radiation sensitivity between pathological and normal tissues. The relative radioresistance of normal brain tissue is attributed to its low mitotic activity and capacity for cellular repair [17]. Another suggested mechanism involves the microvascular damage caused by GKRS; this has been shown in the blood supply of meningiomas and the elimination of the nidus in arteriovenous malformations [17,18,31]. A decrease in blood flow is regarded as an early indicator of response. Another proposed mechanism is the apoptosis of cells, particularly those that are rapidly dividing [17,19,24,25,26,27,28,29,30,31,32,33].

Jumah, F. et al. treated a 13-year-old female patient with GKRS who had been diagnosed with a hypothalamic low-grade glioma through a biopsy. Jumah, F et al. applied a marginal dose of 15 Gy at 50% isodose line for a tumor volume of 2.2 cm^3^. The patient was followed up for 17 years, and the tumoral lesion showed gradual shrinkage until it completely disappeared, and no visual, endocrine, or neurocognitive deficits were present [20,25,26,27,28,29,30,31,32,33,34]. A case series was presented by El-Shehaby, A.M. et al. They treated 22 patients with optic pathway/hypothalamic gliomas for 10 years. Their case series included patients 5 to 43 years old with a median of 16 years of age. The tumor volume ranged from 0.15 to 18.2 cm^3^ with a 3.1 cm^3^ median. The GKRS prescription dose was 8 to 14 Gy with a median of 11.5 Gy. After treatment, twenty patients were followed up for at least 1 year, with a 43-month median. Thirteen patients were diagnosed with hypothalamic/optic chiasm glioma, from which four were diagnosed histopathologically, two were diagnosed with MRI, and seven were diagnosed with MRS. Seven patients developed diabetes insipidus, four patients developed anterior pituitary hormone deficiency, and five patients did not develop any endocrine dysfunction [20,21,35,36,37].

Ge, Y. et al. performed a study that involved 52 patients with optic pathway gliomas who underwent GKRS from 1997 to 2020 with a median age of 13.8 years and a female predominance. Only 6 patients had a hypothalamic/optic chiasm tumor. The mean of the prescription marginal dose was 66.6 Gy and it varied from 26.7 to 126.0 Gy. All patients were followed up for a median of 39 months. One patient with a hypothalamic/optic chiasm tumor required surgical resection, and the remaining five required a second round of GKRS. Ge, Y et al. concluded that GKRS is a safe and effective treatment for treating optic pathway gliomas. Likewise, it can be an option as an initial treatment for children diagnosed with optic pathway gliomas with rapid tumor progression or visual acuity impairment [20,22,38,39].

Somaza, S.C. et al. reported a study in which they used GKRS for treating pilocytic astrocytomas in children. However, their cases only included one hypothalamic case. The range of radiosurgery dose to the tumor margin was 12 to 18 Gy, with a 15 Gy mean. The follow-up time after GKRS was 13 months. Despite treatment, the tumor volume did not decrease, but it remained stable. The patient did not course with complications [23].

Table 4 summarizes the articles by authors who have treated optic pathway/hypothalamic gliomas with radiosurgery.

Liang, C.L. et al. treated two children diagnosed with optic gliomas confirmed as pilocytic astrocytomas. The GKRS plan was 11 Gy to the 50% isodose line for the optic chiasm glioma and 15 Gy to the 50% isodose line for the right optic nerve glioma. Both patients were followed up for 55 to 60 months, and in both cases, near-total disappearance was achieved. Endocrine dysfunction was never present four. Even though these two cases did not involve the hypothalamus, they provide evidence that GKRS is an option for the management of pilocytic astrocytomas.

### 4.4. Hypothalamic Glioma Managed with Surgery

The morbidity and mortality that the surgery entails for the management of these hypothalamic tumoral lesions has been a challenge for the neurosurgeon since it involves the suprasellar region, which includes important anatomical structures such as the optic chiasm, the carotid artery (intracranial portion), the ophthalmic artery, the supraclinoid carotid artery, the posterior communicating arteries, the anterior cerebral artery, and the perforators [24,25,26,27,28,29]. Therefore, managing these lesions without a biopsy might be indicated on the suspicion of pilocytic astrocytoma with less invasive methods such as radiosurgery [25,26].

Godden J. et al. reported a series of 21 patients with optic pathway/hypothalamic gliomas who underwent surgery and reported no visual alterations after surgery. However, they reported one case with transient left hemiparesis that improved with postoperative rehabilitation, and two cases that were complicated with postoperative infection and had to return to the operating room. Likewise, they reported five cases of diabetes insipidus, of which three were permanent. Lastly, they reported an intratumoral hemorrhage biopsy after an endoscopic biopsy, which was resolved with the placement of an external ventricular drainage; unfortunately, the patient died due to fungal ventriculitis [27,28,29,30,31,32].

Pilocytic astrocytomas are typically managed with gross total resection; however, the use of gamma knife radiosurgery (GKRS) has been explored in cases involving brainstem gliomas, optic pathway hypothalamic gliomas, and residual tumors in challenging locations such as the peduncles. While a limited number of studies suggest the potential efficacy of GKRS for low-grade gliomas, the absence of long-term follow-up data restricts a comprehensive evaluation of its effectiveness.

Heppner et al. conducted a retrospective review of 49 patients treated with GKRS between 1989 and 2003, with a median follow-up of 63 months. Their findings demonstrated a median clinical progression-free survival of 44 months and a median radiological progression-free survival of 37 months. At five years, the radiological progression-free survival rate was 37%, while the clinical progression-free survival rate was 41%. Tumor progression resulted in mortality in 14% of cases, and 29% of patients achieved complete radiological remission. The complication rate was minimal, with only 8% of patients experiencing adverse events, of which the majority were temporary or manageable, though one patient developed a permanent neurological deficit.

Despite GKRS’s favorable safety profile, several limitations must be considered. First, GKRS can induce radiation-related changes, such as converting non-enhancing tumors into contrast-enhancing ones within the first 12 months post-treatment, likely due to radiation effects on the blood–brain barrier [33,34,35]. Another reported complication is cystic tumor degeneration. Additionally, the study’s lack of long-term follow-up data and reliance on retrospective analysis limit the generalizability of these findings.

In addition, while GKRS offers a low rate of complications and may provide an alternative treatment for pilocytic astrocytomas and low-grade gliomas in eloquent brain regions or for post-surgical remnants, its long-term efficacy remains unclear. The potential for post-treatment radiological changes and the risk of cystic degeneration highlights the need for further investigation with extended follow-up to better define the role of GKRS in the management of these tumors [20,36,37,38].

## 5. Conclusions

Hypothalamic gliomas, given their critical and eloquent anatomical location, demand a highly individualized and nuanced therapeutic approach to achieve optimal outcomes. These tumors are situated in a region densely populated with vital neurovascular structures, including the optic apparatus, hypothalamus, pituitary gland, and brainstem, making surgical interventions inherently high risk. The most prevalent subtype, pilocytic astrocytoma, presents a unique opportunity for intervention with minimally invasive modalities such as gamma knife radiosurgery (GKRS), particularly in patients where the anatomical complexity of the lesion limits the feasibility of open resection.

Radiosurgery, is a minimally invasive procedure and well suited for managing hypothalamic gliomas. It allows for targeted high-dose radiation delivery to the tumor while sparing adjacent critical structures, thus reducing the risks of post-surgical morbidity and mortality. In cases where clinical and radiological findings suggest tumor progression or recurrence, radiosurgery serves as a primary treatment option due to its ability to achieve effective local disease control with minimal invasiveness. This is especially relevant for hypothalamic lesions, where preserving neurological integrity is paramount.

The application of GKRS offers several advantages over conventional surgical approaches. By providing a non-invasive alternative, radiosurgery circumvents many of the potential complications associated with craniotomy, including infection, hemorrhage, and iatrogenic neurological injury. For patients with tumors in eloquent regions of the brain, such as the hypothalamus, these risks are particularly pronounced. GKRS preserves critical functions and, as demonstrated in this case, significantly improves the patient’s quality of life. The ability to achieve local tumor control without compromising surrounding structures is essential for maintaining neuroendocrine function and overall neurological stability in this delicate area.

This case underscores the efficacy of GKRS as a therapeutic option for hypothalamic gliomas, especially in scenarios where conventional surgical interventions are associated with prohibitive risks. The precision of radiosurgical targeting, combined with the ability to spare surrounding tissues, highlights the role of GKRS in neuro-oncology. As a minimally invasive modality, it allows for effective management of complex lesions in challenging anatomical regions, ensuring patient safety while maximizing treatment efficacy.

Ultimately, GKRS for hypothalamic gliomas exemplifies the growing trend toward precision medicine in neuro-oncology. By tailoring treatment approaches to the unique anatomical and clinical characteristics of each case, radiosurgery provides an optimal balance between tumor control and the preservation of neurological function. This treatment paradigm not only offers superior outcomes but also redefines the standard of care for patients with these complex and challenging tumors.

## Figures and Tables

**Figure 1 jpm-14-01108-f001:**
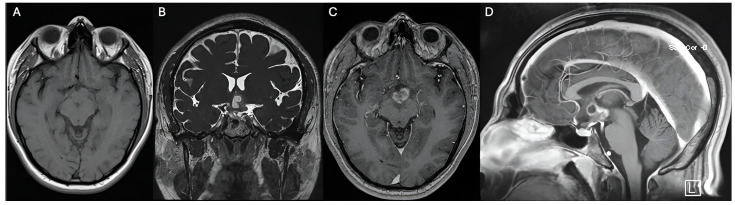
Preoperative MRI shows a hypothalamic lesion that is isointense on T1 (**A**), hyperintense on T2 (**B**), partially enhances contrast (**C**), and has a small cystic component superiorly (**D**). No evidence of calcifications. The hypothalamic lesion has an approximate volume of 8.03 cm^3^.

**Figure 2 jpm-14-01108-f002:**
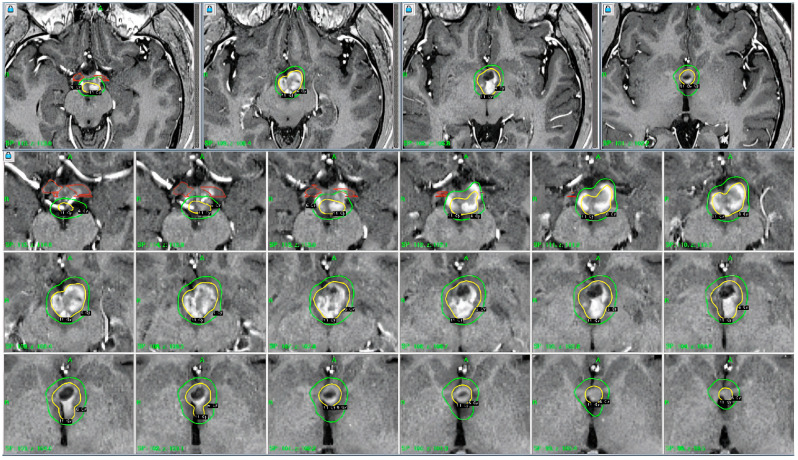
The patient underwent placement of the Leksell stereotactic head frame without difficulty. This was followed by obtaining a high-resolution stereotactic MRI scan of the brain. Contrast-enhanced T1-weighted axial images were obtained at 2 mm intervals. The images were transferred into the treatment planning workstation and reviewed. The previously identified enhanced lesions involving the hypothalamus were observed. Operative procedure: the team proceeded with outlining the visible lesion and developing a treatment plan. A single dose calculation matrix was used. The hypothalamic lesion was treated using 9 shots with 4 and 8 mm collimator helmets. The volume target was 2.6 cm^3^, and the prescription isodose was 11 Gy. White to the 50% isodose line. Note: 100% of the tumor received the prescription isodose. Brain stem < 11 Gy, left optic nerve < 7.8 Gy, and right optic nerve < 5.1 Gy.

**Figure 3 jpm-14-01108-f003:**
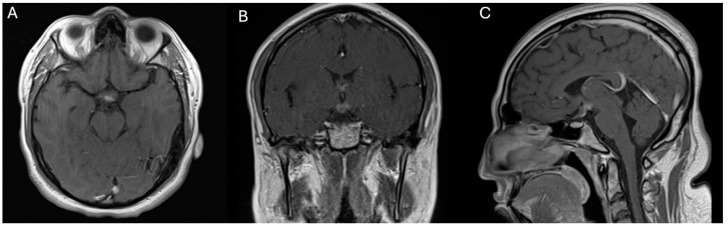
Scans illustrating the 3-month follow-up MRI after gamma knife radiosurgery (GKRS) and revealing a significant reduction in the size of the hypothalamic lesion. The images show minimal contrast enhancement compared to previous imaging studies. The lesion’s volume decreased to approximately 2.19 cm^3^, as visualized in the axial (**A**), coronal (**B**), and sagittal (**C**) MRI sequences.

**Table 1 jpm-14-01108-t001:** Treatment summary.

Target	Shots	Prescription	100% [Gy]	Max [Gy]	X [mm]	Y [mm]	Z [mm]	Grid [mm]
A: Glioma	9	11.0 Gy @ 50%	22.0	22.0	96.5	108.5	107.0	1.0

Total number of shots: 9; beam-on time: 109.6 min.

**Table 2 jpm-14-01108-t002:** First dose.

Run 1 (90 Degrees)													
Run–Step	Shot	X [mm]	Y [mm]	Z [mm]	Collimator [Sector 1–8]								Time [min]
1–1	A6	96.1	111.9	101.3	8	8	8	8	8	8	8	8	8.89
1–2	A5	97.9	114.3	105.7	8	8	B	B	8	8	8	8	11.94
1–3	A8	103.1	114.6	109.5	4	4	B	4	4	4	4	B	9.53
1–4	A9	103.9	111.3	110.3	4	4	4	B	4	B	B	4	10.97
1–5	A2	96.4	101.2	105.6	8	8	8	B	B	B	8	8	14.12
1–6	A1	94.7	103.9	112.1	8	8	8	B	B	B	8	8	12.83

**Table 3 jpm-14-01108-t003:** Second dose.

Run 2 (110 Degrees)													
Run–Step	Shot	X [mm]	Y [mm]	Z [mm]	Collimator [Sector 1–8]								Time [min]
2–1	A3	90.9	109.8	107.7	8	B	B	8	8	8	B	8	14.85
2–2	A4	101.7	109.8	107.7	8	8	B	8	8	8	B	8	12.25
2–3	A7	100.9	103.7	111.9	4	4	4	B	B	B	4	4	14.20

**Table 4 jpm-14-01108-t004:** Patients treated with radiosurgery.

Authors	Article	Year	Number of Patients with Hypothalamic Glioma	Patients Age	Side Effects
Jumah, F. et al. [20]	Gamma Knife Radiosurgery in the Management of Hypothalamic Glioma: A Case Report with Long-Term Follow-up	2022	1	13 years old	None
El-Shehaby, A.M. et al. [21]	Single-session Gamma Knife radiosurgery for optic pathway/hypothalamic gliomas	2016	13	5 to 43 years old, median of 16 years of age	Vision loss in 2 cases DI in 7 cases patients APD in 4 cases
Ge, Y. et al. [22]	A Single-Center Treatment Experience of Gamma Knife Radiosurgery for Optic Pathway Glioma	2022	6	2 to 53 years old, median of 13.8 years of age	Visual acuity preservation was 92%, 84%, and 77% at 1, 3, and 5 years
Somaza, S.C., et al. [23]	Early Outcomes after Stereotactic Radiosurgery for Growing Pilocytic Astrocytomas in Children	1996	1	18 years old	None

## Data Availability

The raw data supporting the conclusions of this article will be made available by the authors on request.

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
