# Peer review of "Radiosurgery for Hypothalamic Gliomas: A Case Report and Clinical Guidelines Form a Neurosurgical Center of Excellence"

_jpm, 2024, doi:10.3390/jpm14111108_

Round 1

Reviewer 1 Report

Comments and Suggestions for Authors

I have read the manuscript “Radiosurgery for Hypothalamic Gliomas: Case Report and 2 Clinical Guidelines form a Neurosurgical Center of Excellence” with interest. It is well presented study, structured accordingly to the standards of such paper.

However, as a surgeon I have some thoughts I would like to share. I the country where I practice it is absolutely outside any medical protocol to conduct radiological treatment on a lesion without a biopsy, since image study data, even the most advanced ones cannot be sufficient for a diagnosis. Treating a benign, demarcated lesion with radiation therapy is also medically questionable. The fact that the lesion showed reduction in size for a 3-month period also raises questions to its histology. And when trying to avoid complication of surgery for a biologically indolent tumor, you should discuss the option to threat endocrinological symptoms and follow up. Nevertheless, I read that the treatment and the study are approved by your ethical committee. Therefore, for opening a debate on a questionable treatment, the paper has a scientific value and should be published because of it.  

Author Response

Dear Reviewers,
We would like to express our sincere gratitude for your valuable feedback on our manuscript. We have
carefully considered all your comments and made the necessary revisions. Below is an outline of the
changes we have made to address your suggestions:
1. Author Information: The authors' full names and affiliations have been updated to include both
first and last names without degrees. The corresponding author has also been clearly identified,
and the affiliations have been completed as per the journal’s guidelines.
2. Introduction: A statement has been added to the purpose of the study, explicitly expressing our
intention to compare the case presentation with data identified in the literature, as requested.
3. Results:
o Lines 97-98: Normal values have been included for the laboratory data.
o Line 98: Figure 1 has been appropriately referenced at the end of the statement.
Similarly, Figure 2 has been attached in the corresponding location within the text.
o Line 116: The treatment regimen used to stabilize diabetes insipidus has been described
in detail.
o Lines 129-132: This statement has been moved to the discussion section, as it was more
suitable for that context.
o Line 138: The unit "cm3" has been corrected throughout the manuscript.
o Tables: The tables have now been attached directly within the text, and the relationship
between paragraph 155-162 and Table 1 has been clarified.

4. Discussions:
o Line 224: The typo "minimally" has been corrected.
o Table 4: Table 4 has been attached within the text at the appropriate location.
o Line 301: The statement "perforators 24" has been clarified.
o Prognostic Factors: A new subsection has been added to discuss prognostic factors
identified in low-grade hypothalamic gliomas, with a comparison to low-grade
supratentorial gliomas, as suggested. We have included the reference to the DOI:
10.3390/curroncol29100576.

5. Additional Sections:
o We have now included updated references for 2020-2024 and sections on Author
Contributions, Funding, Institutional Review Board Statement, Informed Consent
Statement, Data Availability Statement, and Conflicts of Interest at the end of the
manuscript, in accordance with the journal’s requirements.

We hope these revisions address your concerns, and we appreciate the opportunity to improve our
manuscript based on your constructive feedback.
Thank you again for your time and effort in reviewing our paper. We look forward to your further
comments and suggestions.

Reviewer 2 Report

Comments and Suggestions for Authors

Only the first and last names of the authors are written, not the degrees. The corresponding author is not known. Affiliations are incomplete.

Introduction:

Add to the purpose of the study a statement expressing your intention to compare the case presentation with the data identified in the literature.

Results:

Lines 97-98 – add the normal values

Line 98 – add Figure 1 to the end of the statement. Proceed in the same way and appropriately attach figure 2 in the text.

Line 116 – what treatment regimen was used to stabilize diabetes insipidus?

Lines 129-132 – The statement is suitable for discussions, not results.

Line 138 – cm3 instead of cm3. Proceed in the same way until the end of the text.

Attach the tables in the text. It is not clear if paragraph 155-162 should be attached to table 1 or if it is part of the results text.

Discussions:

Line 224 – minimally5 ?

Add table 4 in the text.

Line 301 - perforators 24 ?

Add a short subsection discussing prognostic factors identified in low-grade hypothalamic gliomas and make a comparison with prognostic factors in low-grade supratentorial gliomas (DOI: 10.3390/curroncol29100576)

Add at the end: Author Contributions, Funding, Institutional Review Board Statement, Informed Consent Statement, Data Availability Statement, Conflicts of Interest

Reviewer 3 Report

Comments and Suggestions for Authors

I would like to congratulate the authors for their work. The main aim of their study was to investigate the efficacy of radiosurgery as a treatment modality for hypothalamic gliomas.  The study includes a case report and a literature review. The methodology is clearly explained. The manuscript includes helpful figures and tables and the references are relevant and up to date. The manuscript provides concomitant information for the management surgical vs radiosurgical for the hypothalamic gliomas. The authors concluded that gamma knife radiosurgery is a valid alternative treatment option for the hypothalamic gliomas and spares the surgical complications when approaching this eloquent region. Overall, I think that the study provides useful information for the physicians dealing with hypothalamic glioma. The radiosurgery could be considered as the primary line of treatment in selected of hypothalamic gliomas.
